# Academic Performance during the COVID-19 Pandemic and Its Relationship with Demographic Factors and Alcohol Consumption in College Students

**DOI:** 10.3390/ijerph19010365

**Published:** 2021-12-30

**Authors:** Julio César Vargas-Ramos, Claudia Lerma, Rebeca María Elena Guzmán-Saldaña, Abel Lerma, Lilian Elizabeth Bosques-Brugada, Claudia Margarita González-Fragoso

**Affiliations:** 1Institute of Health Sciences, Universidad Autónoma del Estado de Hidalgo, San Agustín Tlaxiaca 42160, Mexico; julio_ramos@uaeh.edu.mx (J.C.V.-R.); abel_lerma@uaeh.edu.mx (A.L.); lilian_bosques@uaeh.edu.mx (L.E.B.-B.); claudia_gonzalez10101@uaeh.edu.mx (C.M.G.-F.); 2Departamento de Instrumentación Electromecánica, Instituto Nacional de Cardiología Ignacio Chávez, Mexico City 14080, Mexico; dr.claudialerma@gmail.com

**Keywords:** academic performance, college students, COVID-19, alcohol consumption

## Abstract

The COVID-19 pandemic has caused many changes in the education sector worldwide, and school curricula have had to adapt to a non-face-to-face modality. However, international studies have concluded that this modality has affected the academic performance of students. The present study aimed to compare the academic performance of a sample of college students from before the start of quarantine with their current performance, and to test whether various demographic factors influenced these changes in conjunction with alcohol consumption. With a non-experimental, comparative and longitudinal design, we applied an ad hoc questionnaire, in conjunction with the AUDIT questionnaire, in a sample of college students (*n* = 341), and we also obtained data of academic average and failed subjects. The demographic factors that influenced academic performance were sex (*p* < 0.01), age (*p* < 0.01) and alcohol consumption (*p* = 0.001). Most students showed an improvement in their academic average during the quarantine period. Women without failed subjects and low-risk alcohol consumption obtained a better average in this period. In conclusion sex, age and alcohol consumption level were factors associated with academic performance during the quarantine period due to the COVID-19 pandemic; and women had a higher academic average than men did.

## 1. Introduction

### 1.1. Impact of the Coronavirus Disease 2019 (COVID-19) Pandemic in Education

Since the start of the pandemic caused by COVID-19, educational institutions around the world had to adopt a non-face-to-face modality to continue with educational plans, seeking the use of different technologies to allow synchronous and asynchronous communication between students and teachers [1]. Different resources have been taken advantage of such as educational platforms, social networks or instant messaging and video conferencing applications. Although classes have not been interrupted, there is an interest in studying how this contingency period has affected the students, added to the effects on mental health that the general population presents due to fear of contagion, confinement, and economic difficulties [2].

In some studies, a high level of anxiety and stress has been observed in students, caused by the lack of accessibility to technology or the internet, the inefficiency of educational institutions to adopt this modality, and the fear of losing the school year [3,4,5]. These consequences are increased in those students who do not have enough financial and material resources to be able to take their classes under this modality [6]. In addition to this, the students perceive that under this modality, a greater effort is required on their part and that the teachers do not have enough skills to teach using this kind of technology [7].

### 1.2. Academic Performance and Known Factors Affecting It

Academic performance refers to the level of knowledge, skills, and competencies that a student has acquired in the educational field, which is often evaluated with the grades obtained in the subjects that make up the study plan. However, it is a complex concept to study because it can be assessed in different ways, not only with the academic average obtained by the student [8].

Several factors affect negatively the performance of college students. For example, a high anxiety level, depressive symptoms, a negative attitude of the student toward the school or study [9], excessive internet use, poor sleep quality [10], insufficient learning strategies [11], and alcohol or other addictive substances consumption [12,13].

### 1.3. Alcohol Consumption and Academic Performance in College Students

College students present a higher prevalence of alcohol consumption, compared to other populations [14]. This issue has different causes, such as the attitude of students towards consumption, misinformation about alcohol consequences, work overload, lack of parental supervision, family history of substance abuse, pressure from their friends or their couple to consume, and the availability of alcohol in their environment [15,16,17].

Binge alcohol consumption in college students is related to other risky behaviors, such as driving while intoxicated, unprotected sexual contact, illegal drugs consumption, and physical or sexual violence against other people [18], also it is related to affects on physical and psychological health [19].

Previous studies concluded that alcohol consumption is associated with a decrease in college students’ academic performance, which translates into a high failure rate, lower academic average, truancy, and dropout. These consequences have some explanations, firstly in terms of the amount of time that students spend consuming alcohol. It is common for college students to consume alcohol during the week, so the time spent studying is diminished. Also, some immediately physical consequences of consumption like hangovers and mental distress affect students’ performance in all of their daily activities. Finally, the appearance of chronic cognitive issues when the consumption turns into an addiction makes any student activity harder [12,13,20,21].

### 1.4. Academic Performance and the COVID-19 Pandemic

Most of the studies concluded that the students prefer face-to-face activities, and present a negative attitude and low motivation towards virtual education [22,23]. Theoretically, this student perception and attitude can affect negatively the academic performance [3,4,6]; nevertheless, not all the students show a decrease in their academic performance, and some even seem to benefit from this modality [24].

Therefore, it is necessary to know which factors influence changes in the academic performance of students during the pandemic, and if these changes reflect an improvement or a decrease in the academic average. The aims of this study were: (1) to identify the demographic factors that are related to the academic performance of a college students’ sample, (2) to assess whether the alcohol consumption influences the academic performance, and (3) to compare their performance before confinement by COVID-19, and a year later.

## 2. Materials and Methods

### 2.1. Study Design

To meet the first two aims of this research, we used a non-experimental, comparative, prospective and transversal design; no variables were intentionally manipulated and all data were obtained using questionnaires in a single moment. For the third aim we used a longitudinal design, since the academic performance was obtained at two moments, with 1 year between each evaluation; again, no variables were manipulated.

### 2.2. Participants

The sampling was non-probabilistic and for convenience, and the participants were college students of the Degree in Psychology (*n* = 341, 73.3% women, age = 19.8 ± 2.1), from a public university in the State of Hidalgo in Mexico.

### 2.3. Evaluation Instruments

To obtain the sociodemographic variables, we used an ad hoc questionnaire prepared in which the participants were questioned about their age, sex (woman/man), marital status (single/married/free will), who they live with (parents/relatives/friends/alone/partner), the shift in which they study (morning/evening), and the city or community which they come from, in case of being foreign students. With these data, the type of locality from which the students come from was identified (metropolitan/urban zone/suburban area/rural area), according to the classification established in the National Urban System [25].

Regarding academic performance, the data referring to the academic average and the number of failed subjects of the participants were used in two moments: before the start of the quarantine, and a year later, during the quarantine. The academic authorities of the institution, where the research was carried out, provided this data.

For the assessment of alcohol consumption, we used the Alcohol Use Disorders Identification Test (AUDIT) [26], which consists of 10 items with four Likert-type response options, and provides a score that allows classifying the participant’s level of alcohol consumption. A score between one and seven is equal to a “Low-risk consumption”, a score between eight and 15 is equal to a “High-risk consumption”, and a score equal to or greater than 16 is equal to a “Dependence-risk consumption”. The AUDIT was validated in the Mexican population; with 0.88 and 0.81 Cronbach’s alpha score [27].

### 2.4. Statistical Analysis

We use the Chi-square test to identify the qualitative variables that could be related to the failure history of the participants. For the quantitative variables that presented a normal distribution, a Student’s *t*-test was performed comparing the scores of the groups, and for the variables that did not present a normal distribution, we used the Mann–Whitney-*U* test. The data are described as mean ± standard deviation, absolute (relative) frequency, or median (25th–75th percentiles). An ANOVA with a Bonferroni post hoc test was used to compare the academic average among the participants, divided into groups according to their sex, level of alcohol consumption, history of failure, and the evaluation period (before or during the pandemic). To perform all the statistical analysis, we used the IBM SPSS software for Windows, version 21.0(IBM, SPSS Inc., Chicago, Illinois USA). A *p*-value < 0.05 was considered statistically significant.

## 3. Results

### 3.1. Descriptive Results and Demographic Factors Related to Academic Performance

The study sample consisted of 341 university students, of which 250 (73.3%) were women and 91 (26.7%) were men; they had a mean age = 19.8 (*SD* = 2.1), and 330 (96.8%) reported being single, seven (2.1%) lived in a common-law union, and four (1.2%) were married. About the people they lived with, 226 (66.3%) reported living with their parents, 32 (9.4%) live with other relatives, 23 (6.7%) live with friends, 10 (2.9%) live with their partner, and 50 (14.7%) live alone. Also, 270 (79.2%) come from a metropolitan locality, 26 (7.6%) from an urban locality, 7 (2.1%) from a suburban zone, and 38 (11.1%) from a rural zone. Finally, 202 participants (59.2%) study in the morning shift, and 139 (40.8%) study in the evening shift.

In the alcohol consumption assessment, the student’s sample achieved a median of 3.0 (1.0–6.0) in the AUDIT score; about the level of consumption, 82 students (24%) reported not consuming alcohol, 197 (57.8%) presented a low-risk consumption, 48 (14.1%) presented a high-risk consumption, and 14 (4.1%) presented a dependency-risk consumption level. 

The sample was divided into two groups: (a) those who did not fail a subject before the quarantine (*n* = 281), and (b) those who failed one or more subjects in the same period (*n* = 60).

Table 1 shows the results of Student’s *t*-test, Mann Whitney-*U* test and Chi square tests that showed statistical significance. Compared with students without failed subjects, the group with at least one subject failed was older, with a lower proportion of women, a higher AUDIT score, and a higher proportion of high-risk alcohol consumption.

### 3.2. Comparison between Groups

Figure 1 shows the academic average mean of the students obtained before the quarantine (circles) and during the quarantine (squares), grouped by the academic record of failure, type of consumption, and sex. Students with a record of failing one or more subjects present a significantly lower academic average in comparison with the groups of students that have not failed any subject (*p* < 0.05). When we compared the groups according to the alcohol consumption, there is only a significant difference in women that have not failed any subject. In this case, the women with a low-risk consumption level has a higher academic average during the quarantine, compared with the women with high-risk alcohol consumption in this same period (*F* = 4.11, *p* = 0.04).

When we compared the groups according to sex, we observed that women with a record of not failing and low-risk consumption had a significantly higher academic average, in comparison with men with the same features, before the quarantine (*F* = 5.06, *p* = 0.02), and during this period (*F* = 5.38, *p* = 0.02). Also, women with failure records and high-risk alcohol consumption has a significantly higher academic average during the quarantine, in comparison with men with the same features and the same period of evaluation (*F* = 4.322, *p* = 0.04).

Finally, when we compare the academic average of all the groups before the quarantine against the average obtained during this one, we found a significant improvement in the men with a record of not failing and low-risk consumption (*F* = 11.264, *p* = 0.001), and in men with the same record and high-risk consumption (*F* = 5.221, *p* = 0.02), Figure 1. The women with no failed record and low-risk alcohol consumption also had a significant improvement in their academic average during the quarantine (*F* = 43.688, *p* < 0.001).

## 4. Discussion

This research explored the consequences of the COVID-19 pandemic in Mexico, comparing the academic average of a sample of psychology degree students, before the beginning of the quarantine and during this period, considering factors like alcohol consumption and demographic features.

### 4.1. Demographic Variables Related to Academic Performance

Among the demographic variables related to academic performance, we highlight the sex distribution of the students. Generally, women had better academic performance compared to men. This difference may be caused by several factors; for example, it has been reported that women present a better attitude to the ICT, computers, and internet use [28], better time management, effort, and learning strategies, compared with men [11]. Social and cultural aspects can also influence this difference, parents educate differently depending on the sex of their child and this education can influence the student’s learning skills, habits, and attitude towards school. Also these differences in the treatment of the student, depending on their sex, are also responded to by teachers at school, so this can affect the academic performance of the students [29]. Finally, another factor that can explain this difference depending the sex, is alcohol consumption; it is known that men present a higher prevalence of binge consumption, and this could affect negatively their academic performance in comparison with women, even though in recent years, this difference between men and women has decreased [30].

Age was another demographic variable that showed a significant relationship with the academic performance. We found the students that failed one or more subjects were older than the students that did not fail. This result agreed with some studies that concluded that the academic performance of the students was negatively affected at older ages. This relationship could be explained by the fact that older students began to carry out other types of activities such as work or going out with friends, so the time invested in school was reduced [31].

Coupled with this, it has been observed that academic performance is associated with the socioeconomic level of the student and their family, as well as the availability of technology and the internet [6]. In our results, we observed that the type of community (metropolitan, urban, suburban and rural) in which the students lived was not related to their academic performance. This could be explained by the fact that in this sample of students there is not such a disparity between those who live in rural communities and those who live in urban cities.

### 4.2. Alcohol Consumption and Academic Performance

Related to the alcohol consumption, in our results we observed that the students with high-risk levels also had a lower academic average and more failed subjects, in comparison with the students with low-risk levels. This agrees with previous studies in which it is concluded that binge alcohol consumption and use of other addictive substances, affect in a negative way the academic performance [12,13,20].

Despite this, students showed an improvement in their academic average during the quarantine, which could indicate a decrease in the alcohol consumption of students. Although there are no data on the students’ consumption in this period, it could be a viable explanation, mainly due to the supervision of their parents [32], the implementation of the Prohibition Law at national level during the first months of the contingency [33], the closing of bars and clubs [34], and the inability to meet with colleagues and friends [17].

### 4.3. The Pandemic and Academic Performance

Despite the fact that some studies conclude that the pandemic has negatively affected students [3,4,6], in our results we found an improvement in the academic performance of the students, comparing their average before the beginning of the quarantine and their average obtained 1 year later. There is evidence of certain benefits caused by the pandemic, such as the fact that students have greater interaction with their family, have more parental supervision, and they can perform other activities and hobbies that are less likely when they have to go to school, so these benefits could influence in the improvement of their academic performance [22].

Another explanation of this improvement is related to the educational institutes and how they adapted their curricula to a non-face-to-face modality. If the institution’s authorities bring the necessary support and required tools to their students, they will be motivated to continue with their studies, and their academic performance will not be affected by the pandemic. In the same way, if the teachers are properly trained to teach their classes under this modality, the educational objectives can be achieved [24].

### 4.4. Limitations of the Study

Some limitations of this study were, in the first place, that the sampling of this research was not probabilistic, so these results cannot be generalized to all the population of college students. On the other hand, we evaluated the academic performance using the academic average and the number of failed subjects, and it was decided not to take into account other indicators. Likewise, future studies are required to evaluate other factors that could influence academic performance (for example, the school infrastructure and the student’s family or personal situations). Finally, due to the characteristics of the population and the methodology used, the groups that were compared were unevenly integrated; coupled with this, the distribution of the participants according to their sex was unequal because most of the participants were women, so the number of members of each group and the sample divided by sex could influence some results.

For future studies, we suggest using probabilistic sampling, so the results obtained can be generalized in the population. In addition, to gain more knowledge, future researchers may use a mixed methodology to know the own experience of students and teachers during the teaching of classes under the virtual modality and monitor alcohol consumption throughout the contingency, in conjunction with other types of variables such as anxiety and depressive symptoms.

## 5. Conclusions

In general, students showed an improvement in their academic performance during the confinement caused by the COVID-19 pandemic, compared to the period before this. We concluded that the interaction with family, more parental supervision, the possibility to perform hobbies and other activities in addition to study, and the method used by the educational institute to adapt to a non-face-to-face modality and all the support for the student’s requirements, may have influenced this difference on their academic performance. Of the demographic factors that could be related to students’ performance, only age and sex showed a significant relationship, together with alcohol consumption. The group of students who showed the best academic performance during the quarantine period was that comprising women with no failed subjects and low-risk alcohol consumption. More studies with mixed methodology and probabilistic sampling are required to give continuity to this type of study.

## Figures and Tables

**Figure 1 ijerph-19-00365-f001:**
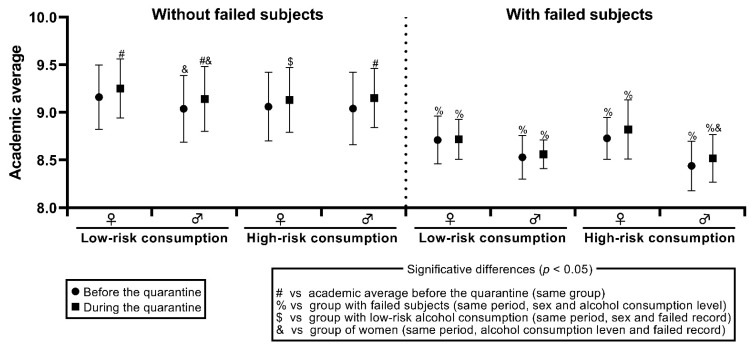
Comparison of the academic average of the students, according to their failed record, alcohol consumption level, sex, and assessment period. Results are shown as main and standard deviation.

**Table 1 ijerph-19-00365-t001:** Descriptive results of the students’ sample of the study.

Variables	Total Sample (*n* = 341)	Has Failed a Subject		Statistics	*p*-Value
		No (*n* = 281)	Yes (*n* = 60)		
Age (years)	19.8 ± 2.1	19.6 ± 2.2	20.4 ± 1.9	*t* = 2.8	0.007
Sex				*X*^2^ = 23.2	<0.001
Women	250 (73.3%)	221 (78.6%)	29 (48.3%)		
Men	91 (26.7%)	60 (21.4%)	31 (51.7%)		
AUDIT score	3 (1.0–6.0)	2 (0.0–5.0)	6 (3.0–9.0)	*U* = 5372.5	<0.001
Alcohol consumption level				*X*^2^ = 16.7	0.001
Null consumption	82 (24.0%)	77 (27.4%)	5 (8.3%)		
Low-risk consumption	197 (57.8%)	161 (57.3%)	36 (60.0%)		
High-risk consumption	48 (14.1%)	35 (12.5%)	13 (21.7%)		
Dependence risk consumption	14 (4.1%)	8 (2.8%)	6 (10.0%)		

Data is shown as mean ± standard deviation, median (percentile 25–percentile 75) or absolute value (percentage). AUDIT = Alcohol Use Disorders Identification Test

## Data Availability

The data presented in this study are available upon request from the corresponding author.

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
