# Peer review of "Academic Performance during the COVID-19 Pandemic and Its Relationship with Demographic Factors and Alcohol Consumption in College Students"

_ijerph, 2021, doi:10.3390/ijerph19010365_

Round 1
Reviewer 1 Report
- Introduction
1) 1.1 Academic performance and known factors affecting it
In previous studies, what were the general characteristics that affected the academic achievement of college students during the COVID-19 epidemic, and how did they affect their academic achievement?
2) 1.2 Alcohol consumption as a risk factor in college students
An explanation of the relationship between alcohol consumption and academic achievement among college students is needed.
- Materials and Methods
3) Please add more information about the sociodemographic variables to the text.
- Results
4) please consider the purpose of the study, then rearrange the table. The purpose of the study and the table are confusing.
5) “On the other hand, there were no differences between the groups, depending on their marital status, the people who lived with them, the type of locality that they come from, and the study’s shift.”: Is this statement necessary?
6) Please check 100% in the table.
7) Table 2. Academic average of the participants, according to their failed record, alcohol consumption level, and sex.
Is the table related to the research purpose?
- Discussion
8) Is the description of age missing?
9) The author should discuss separately the sociodemographic characteristics and drinking variables that can affect the academic achievement of college students.
10) Further explanation is needed for gender differences in academic achievement.
11) What is the reason for the increase in academic achievement of college students during the epidemic compared to before the COVID-19? More than 70% of women participated in this study compared to fewer men, and women were more likely to have higher academic achievement than men. This may have affected the results of this study. It should be added to the Limitation section.
References
12) Many references related to the subject of this study have been published since COVID-19. 2003, 1998, etc. were too long ago.
Author Response
Please see the attachment
Regards.

Reviewer 2 Report
In the introduction there are presented the theoretical elements of the paper at a general level, in terms of alcohol consumption in college students, as well as at a specific level, from the perspective of the pandemic period. It is recommended to elaborate separately the introduction and move the theoretical elements to the next point. The introductory part will highlight the importance of the topic and the reasons behind the implementation of the study.
The research design must be presented separately and it must be argued why the comparative and longitudinal research methodology was used. Participants must be presented separately, mentioning the sampling technique.
Statistical analysis is complex and structured.
Discussions of research results are well elaborated.
The conclusions section is very brief. It is necessary to add some recommendations and solutions.
There is the approval of the ethics commission for the implementation of the research.
Author Response
Please see the attachment
Regards.

Round 2
Reviewer 1 Report
Thank you for accepting the reviewer's comments.